# Impact of Social Support and Social Trust on Public Viral Risk Response: A COVID-19 Survey Study

**DOI:** 10.3390/ijerph17186589

**Published:** 2020-09-10

**Authors:** Eugene Song, Hyun Jung Yoo

**Affiliations:** Department of Consumer Science, Chungbuk National University, Cheongju 28644, Korea; eugenesong@chungbuk.ac.kr

**Keywords:** distancing activity, public health, purchasing activity, response efficacy, sanitation activity, social support, social trust, viral risk

## Abstract

Public health has been under continuous threat worldwide in recent years. This study examined the impact of social support and social trust on the activities and efficacy of the public’s risk response in the case of COVID-19. We conducted an online survey over eight days with 620 Korean adult participants. Data were analyzed using structural equation modelling and K-means cluster analysis. Our results showed that public support had a positive impact on response efficacy, while response efficacy had a positive impact on sanitation, distancing, and purchasing activities. In addition, social support positively moderated the impact of public and individual support on response efficacy, while response efficacy negatively moderated the impact on sanitation activities. These results suggest that, first, amid viral risk, governments should proactively supply tools and information for infection-prevention, and deliver messages that encourage and support infection-prevention activities among the public. Second, when viral risk occurs, governments, along with all other members of society, must engage in aggressive risk response measures. Third, there is a need for risk communication that further emphasizes the importance of personal sanitation activities in the face of viral risk.

## 1. Introduction

In recent years, public health has been under continuous threat worldwide. Among other health threats, the severe acute respiratory syndrome (SARS) epidemic in 2003, the novel influenza virus in 2009, and the Middle East respiratory syndrome (MERS) in 2015 have taken a substantial toll on various societies and their economies [1]. The outbreak of coronavirus disease (COVID-19) in 2020 is viewed as a particularly serious risk, with an enormous global impact, including the closure of national borders and mandatory quarantines. COVID-19 is a respiratory infection that quickly spread around the world since its first outbreak in Wuhan, China. While initially known only as a respiratory infection of unknown etiology, on 9 January 2020, the World Health Organization (WHO) confirmed that the pneumonia was caused by a novel coronavirus [2].

Between 20 January, when the first case of COVID-19 had been confirmed in South Korea, and 16 February 2020, the daily average number of new cases remained at 1.03. However, the number of confirmed cases began to skyrocket after the diagnosis of the 31st patient on 17 February 2020. According to the South Korean central government and the Korea Centers for Disease Control and Prevention (KCDC) report [2], an average of 207.4 additional confirmed cases per day were reported from 17 February 2020, to 13 April 2020. On 29 February 2020, an additional 1128 cases were confirmed to be infected with COVID-19. Due to aggressive management and prevention measures implemented by the KCDC, the rate of additional diagnoses dropped significantly after 14 April 2020, to an average of less than 30 cases per day. However, in June 2020, South Korean authorities announced the potential need for stricter physical measures should the increasing infection rate not subside [3,4].

As COVID-19 is a novel pathogen and, as of August 2020, a vaccine is yet to be developed, South Korean authorities have devoted tremendous efforts to preventing the countrywide spread of COVID-19 by implementing preventive measures such as social distancing, effective handwashing, and coughing etiquette, as well as the careful management of confirmed patients and their close contacts [2]. These measures include the provision of necessary equipment, such as the provision of “public masks,” masks provided by designated pharmacies for a low price in response to the soaring prices of facial masks, and the order of a supply of hand sanitizers by major hospitality facilities; the dissemination of information aimed at overcoming the COVID-19 pandemic, such as information on the current number of confirmed cases and patients’ paths, correct use of masks and handwashing, and social distancing; and the provision of psychological support through advertisement campaigns that support and encourage people’s infection-prevention behaviors [5].

According to Bei et al. [6], a lack of social support at a time of risk has an adverse impact on people’s physical and mental health. Horowitz [7] stated that social support is a protective factor moderating psychological impairment in people who had been exposed to a trauma. One factor that is strongly related to risk control is efficacy [8], and social support plays a crucial role in overcoming the COVID-19 pandemic by influencing efficacy [9]. Another important factor is social trust. Social trust is based on cultural values and refers to decisions about whether to trust or distrust anonymous others (people and institutions) [10]. It provides psychological stability when individuals interact with others and is therefore of great import in increasingly atomistic modern societies [11]. Social trust impacts a person’s overall perception of risk and their psychological response and behavior related to the risk [12,13], as well as effectively promoting risk recovery [14,15].

Viral risk calls for multilateral infection-prevention efforts by the general public residing in the affected and adjacent regions, even if they have not been infected by the virus, and it is necessary to explore the roles of social support and social trust as potential factors of risk response efficacy among members of the public exposed to a viral risk. Our study aimed to (1) develop and test a model to examine the impact of social support on risk response efficacy and activities during the COVID-19 pandemic; and (2) determine how social trust moderates the impact of social support on risk response efficacy and risk response activities.

This study aimed to show how governments must effectively and proactively provide material, information, and psychological support and that social trust must be fostered through active response activities against the virus by all members of society in order to improve the public’s risk response ability and activities. Theoretically, the study findings will be conducive to understanding the public’s behavior in the face of viral risk, and practically, the findings will serve as evidence for developing measures to overcome such risk.

## 2. Theoretical Background and Hypothesis Development

### 2.1. Social Support and Response Efficacy

After a disaster, victims mutually interact with various dimensions of the surrounding environment, and social support is one of the most crucial predictors of these interactions [16]. Social support refers to all positive resources people can obtain through interpersonal relationships and the act of interacting with and receiving help from other people to satisfy one’s social needs [17]. Song [18] described social support as a positive resource acquired while interacting with others, and stated that it motivates people to engage in a particular action by increasing their will to perform that particular behavior.

Scholars have most commonly categorized social support using two classification types: classification by function of support, namely material support, informational support, and emotional support [18,19,20]; and classification by the provider of support, namely individual support provided by personal contacts, including family and friends, and public support provided by government and local communities [18,21,22]. In this study, we classified social support by provider type (individual and public support) and examined its relationship with response efficacy.

Self-efficacy is an important concept related to perceived control over risk. The construct was first introduced in Bandura’s [8] social learning theory and refers to one’s belief in one’s ability and knowledge for performing a particular behavior [23]. Response efficacy is one of the parameters theoretically related to self-efficacy and is theoretically based on the expectancy value theory [24]. People with a high response efficacy have higher positive expectations for outcomes, increasing their likelihood to engage in a particular behavior. Response efficacy encompasses the construct of self-efficacy, which is one’s belief in one’s abilities to perform a particular behavior.

The relationship between social support and response efficacy has been the subject of a number of studies. In general, social support has been studied as a factor that increases health behaviors in patients or older adults. Duncan and McAuley [25] and Resnick et al. [26] reported that social support, including public and individual support, indirectly boosts exercise behaviors through efficacy in older adults. Berkman [27] found that peer and colleague support increases individuals’ willingness to exercise, thereby demonstrating that individual support has a positive effect on efficacy. Wind et al. [28] found that high levels of cognitive social support leads to better mental health outcomes following a disaster. Further, Bashirian et al. [29] reported that social support affects breast self-examination behavior, implying that social support can boost self-efficacy. Social support also has an important impact on risk. Yoo and Joo [30] and Kim et al. [31] revealed that public information support regarding a risk impacts response efficacy: individuals with more knowledge about a risk showed increased willingness to respond to that risk. Further, Lee and Choi [32] reported that people with a high risk response efficacy are generally more interested in their health and are actually well informed of the ways to protect themselves from harm, while people with a low response efficacy lack knowledge on and belief in the recommended methods for addressing a risk. This indicates that knowing how to control a risk, that is, having informational support, increases response efficacy.

In summary, social support can be classified as public support and individual support and could have a positive impact on response efficacy. Thus, we established the following hypotheses:

**Hypothesis 1 (H1)**.
*Public support has a positive impact on response efficacy.*


**Hypothesis 2 (H2)**.
*Individual support has a positive impact on response efficacy.*


### 2.2. Response Efficacy and Activities

Activities are the outcomes of an individual’s final decision based on their pool of information on a risk [33]. If disaster is viewed as a risk, activities can be examined based on people’s risk responses. Neal and Griffin [34] defined safety behaviors as behaviors performed to protect one’s own safety, help others at risk, or create a safe environment. Song [35] described safety behaviors as self-protective behaviors to prevent a harmful event and as the outcome of individuals’ thoughts of ensuring the safety of others and themselves. Safety behaviors during the COVID-19 pandemic are those behaviors demonstrated by the members of the public aimed at protecting themselves and others from the risk of COVID-19.

Korea’s Central Disaster Management Headquarters, Central Disease Control Headquarters [2], recommends practicing good personal hygiene, such as properly wearing a facial mask, washing hands frequently, and covering the mouth and nose when coughing or sneezing, to prevent the spread of COVID-19. Further, to prevent direct contact, the Korean public has been advised to refrain from outside activities as much as possible and to practice physical distancing by maintaining a distance of at least 2 m from each other if they must engage in outside activities. According to Galea et al. [36], the practice of social distancing may be crucial to suppressing the spread of the disease. Preventing COVID-19 infection essentially requires the use of products to help prevent infection, such as face masks, antimicrobial and disinfecting products, and antibacterial soap, encouraging the widespread purchase and consumption of these products [37]. Lee and Yoo [38] classified public response activities into precautionary behaviors, including wearing facial masks and hand hygiene, and distancing behaviors, such as a reduced use of public transportation, avoiding crowded places, and postponing or canceling social events. In sum, public responses to the COVID-19 pandemic can be divided into personal hygiene practices, social distancing practices, and purchase of infection-prevention products.

The relationship between efficacy and infection response behaviors has been examined in a number of recent studies. Lee and Yoo [38] reported that self-efficacy positively affected precautionary and distancing activities early in the pandemic. Cui et al. [39] showed that self-response efficacy had an effect on protective intension in the case of avian influenza A/H7N9. Tuma et al. [40] reported that efficacy related to an infectious disease had a positive impact on parents’ intention to vaccinate their children. In a study investigating the effects of a media campaigns on influenza-prevention behaviors, Lee et al. [41] observed that people’s efficacy related to influenza increased their willingness to comply with prevention recommendations. Chang and Shim [42] also shed light on the fact that efficacy related to infectious diseases such as foot and mouth disease and novel influenza had a positive impact on infection-prevention behavioral intention.

Taken together, response efficacy has a positive impact on infection-prevention behaviors, and we established the following hypotheses:

**Hypothesis 3 (H3)**.
*Response efficacy has a positive impact on sanitation activity in response to a viral risk.*


**Hypothesis 4 (H4)**.
*Response efficacy has a positive impact on distancing activity in response to a viral risk.*


**Hypothesis 5 (H5)**.
*Response efficacy has a positive impact on purchasing activity in response to a viral risk.*


### 2.3. Social Trust

Social trust is derived from trust, a component of social capital [43]. Levi and Stoker [44] stated that social trust is an essential component of social collaboration, and Poortinga and Pidgeon [45] and Mehta et al. [46] also considered trust to be an essential component in interaction pertaining to risk. Social trust can be viewed as an important predictor for overcoming the viral risk of COVID-19, which calls for the cooperation of all members of the society to prevent the spread of infection.

Considering that social trust helps overcome barriers to collective action and effectively promotes resilience and recovery [14], it is also a factor that predicts risk response efficacy and activities. Lui and Mehta [47] reported that trust affects public practice. Rickard et al. [48] also mentioned that interagency trust can enable authorities to coordinate their activities and share information quickly. In particular, social trust helps overcome problems related to collective action that may hinder recovery, and communities with high social trust demonstrate a higher resilience than those with poor social trust [49], suggesting that social trust moderates resilience. The belief that society can complement individuals’ vulnerability to risk has a positive effect on risk response efficacy and actions to overcome risk in victims [16,50,51,52]. Further, Aldrich [49] reported that governments’ resilience is based on social trust and bonding. Based on these findings, we hypothesized that social trust would moderate social support, risk response efficacy, and actions to overcome risk, and established the following hypotheses:

**Hypothesis 6 (H6)**.
*Social trust will moderate the impact of public support on response efficacy.*


**Hypothesis 7 (H7)**.
*Social trust will moderate the impact of individual support on response efficacy.*


**Hypothesis 8 (H8)**.
*Social trust will moderate the impact of response efficacy on sanitation activity.*


**Hypothesis 9 (H9)**.
*Social trust will moderate the impact of response efficacy on distancing activity.*


**Hypothesis 10 (H10)**.
*Social trust will moderate the impact of response efficacy on purchasing activity.*


Freitag and Traunmuller [53] and Uslaner [54] classified trust as specialized and generalized trust. Specialized trust refers to individuals’ trust in people they know personally, such as family, friends, and colleagues, while generalized trust refers to a general and rather abstract level of trust among the general public, including strangers, unspecified groups, and broader society [55]. Habibov and Afandi [56] categorized trust into interpersonal trust and institutional trust, and Park [57] examined private trust, including personal connections, and public trust in public institutions. In sum, social trust can be divided into trust in public organizations, such as the government, trust in the general public, and trust in personal contacts.

## 3. Methods

### 3.1. Data Collection

Study data were collected using a questionnaire administered by a professional online survey company, Macomill Embrain, for eight days (18–25 May 2020). The company has a panel of 1,249,392 adults aged 20 or older, which is equivalent to 3.5% of the South Korean population. The questionnaire was converted to an online-based survey to make online administration easier and was distributed to a total of 7320 panelists via email. One panelist received no more than two response invitations. Prior to beginning the survey, participants were asked whether they voluntarily provided their consent to participate, and only those who indicated their voluntary consent were allowed to participate in the survey. The panelists that completed the online survey received incentives equivalent to about $3. The final response rate was 8.5%. Questionnaire responses collected within a certain time frame were collected and analyzed. A total of 620 participants were selected using the proportionate quota sampling method, which was based on population, gender, age, and area of residence (Table 1). This study received ethics approval from the Research Ethics Committee of Chungbuk National University (CBNU 202006-0109).

### 3.2. Measures

In this study, the following six items were developed: public support (PS), individual support (IS), response efficacy (RE), sanitation activity (SA), distancing activity (DA), infection-prevention product purchasing activity (PA), and social trust (ST) (Table 2).

#### 3.2.1. Social Support

In this study, we classified social support into public support and individual support based on the source of support, with reference to Song [18] and Rho and Mo [21], and developed hypotheses accordingly. Further, social support was classified into emotional support, informational support, and material support based on the function of support, with reference to Thompson et al. [19], Park [20], and Song [18]. All items were measured on a five-point Likert scale, with 1 = “strongly disagree,” and 5 = “strongly agree.”

#### 3.2.2. Risk Response Efficacy

Response efficacy was measured using a modified version of the items for measuring response efficacy developed by Floyd et al. [58] They defined response efficacy as an individual’s belief in their ability to engage in activities to protect their own health in the face of risk using the items listed in Table 2. All the items were measured on a five-point Likert scale (1 = “strongly disagree”; 5 = “strongly agree”).

#### 3.2.3. Risk Response Activities

As previously discussed, COVID-19 response activities can be divided into infection-prevention activities and infection-prevention product purchasing activities, with infection-prevention activities further divided into sanitary activities and distancing activities [2,36,37,38]. We developed a scale to measure sanitation activity using three items (Table 2) measured on a five-point Likert scale (1 = “strongly disagree”; 5 = “strongly agree”).

#### 3.2.4. Social Trust

With reference to Freitag and Traunmuller [53], Uslaner [54], Seo [55], Habibov and Afandi [56], and Park [57], we classified social trust based on public trust (trust in the government), unspecified public trust (trust in the general public), and individual trust (trust in personal acquaintances) and measured perceived trust using the items listed in Table 2. All the items were measured on a five-point Likert scale (1 = “strongly disagree”; 5 = “strongly agree”).

### 3.3. Analysis

Analyses were done by transferring the data to SPSS 21.0 and AMOS 21.0 software. Participants’ demographic characteristics were analyzed using frequency analysis. The baseline measures of each parameter were analyzed with descriptive statistics, namely mean and standard deviation. The validity of the scale was tested with confirmatory factor analysis (CFA) and reliability analysis. This study determines the fitness of the model through the following criteria: the Chi-square minimum/degree of freedom (CMIN/df), the root mean-squared residual (RMR), the root-mean-squared error of approximation (RMSEA), the goodness-of-fit index (GFI), the adjusted-goodness-of-fit index (AGFI), and the comparative-fit index (CFI). For a model with acceptable quality, the following threshold values are recommended: CMIN/df ≤ 5, RMR ≤ 0.05, RMSEA ≤ 0.08, GFI ≤ 0.9, AGFI ≥ 0.9, CFI ≤ 0.9 [59,60,61,62]. Further, the reliability of the scale was tested by comparing the standardized and variance estimates computed from CFA, composite reliability (CR), average variance extracted (AVE), square roots of AVEs, and Cronbach’s α. For a model with acceptable quality, the following threshold values are recommended: CR ≥ 0.7, AVE ≥ 0.5, Cronbach’s α ≥ 0.7, square roots of AVEs ≥ of the interconstruct correlation coefficient [63,64].

H1–H5 were tested using structural equation modeling (SEM). H6–H10 were tested with reference to the method of testing moderating effects suggested by Harish Sujan and Weitz [64], Woo [65], and Lee and Lim [66]. First, K-means cluster analysis was performed to cluster groups according to the mean values of the moderators. Next, multiple group SEM was performed by applying constraints for each path and the difference in the standardized path coefficients between two corresponding groups in the unconstrained model. The path coefficients were considered statistically different when the absolute value of the composite reliability between standardized path coefficients of each group was 1.965 or higher, and a moderating effect was confirmed upon fulfilling this criterion.

### 3.4. Model

We established the following hypotheses and study model to examine the impact of social support on risk response efficacy and activities during the COVID-19 pandemic. Ten study hypotheses were developed to examine the relationships among PS, IS, RE, SA, DA, PA, and ST, and the structure is illustrated in Figure 1.

## 4. Results

### 4.1. Measurement Model

To determine whether the measurement model was suitable, we assessed its content validity, convergent validity, and discriminant validity. First, content validity was tested by developing constructs and measurement items based on the literature and conducting a pilot test. Second, convergent validity was tested by examining the fit indices computed in the CFA. The results were χ^2^/df = 2.627, RMR = 0.038, RMSEA = 0.051, AGFI = 0.916, and CFI = 0.970, confirming a good model fit. In addition, standardized estimate, Cronbach’s α, CR, and AVE were examined. As shown in Table 3, factor loading, Cronbach’s α, and CRs for each construct were all over the acceptable threshold of 0.7, and AVEs for each item were all higher than the acceptable threshold of 0.5 [67]. Therefore, convergent validity is supported. Discriminant validity was tested using interconstruct correlation coefficients. The square roots of AVEs for each underlined construct are higher than the other values [68].

### 4.2. Structural Model 1

The fit indices were χ^2^ = 309.592***, DF = 107, CMIN/DF = 2.893, RMR = 0.059, RMSEA = 0.055, GFI = 0.943, AGFI = 0.919, and CFI = 0.969, based on which the model was determined to have an acceptable fit. Figure 2 illustrates the structural model, where a solid line represents the model supporting the study hypothesis and a dotted line represents the rejection of the study hypothesis. PS had a positive effect on RE (β = 0.268, *p* < 0.000), supporting H1. However, IS did not have a statistically significant effect on RE, not supporting H2. Next, RE had a statistically significant effect on SA (β = 0.639, *p* < 0.000), DA (β = 0.649, *p* < 0.000), and PA (β = 0.212, *p* < 0.000), thereby supporting H3, H4, and H5.

### 4.3. Structural Model 2

K-means cluster analysis was performed to divide the group with high ST from that with low ST. The high ST group consisted of 507 (81.8%) people, with a mean ST score of 4.381 (±0.455). The low ST group consisted of 113 (18.2%) people and had a mean ST score of 2.844 (±0.560).

The moderating effect of ST was analyzed with multiple group SEM for each group. The fit indices were χ^2^ = 438.024***, DF = 214, CMIN/DF = 2.047, RMR = 0.052, RMSEA = 0.041, GFI = 0.935, AGFI = 0.903, and CFI = 0.961, based on which the model was deemed acceptable. Figure 3 shows a diagram of the analysis of the moderating effects of ST, where a solid line represents the presence of a moderating effect by ST, that is, supporting the study hypothesis, and a dotted line represents the absence of a moderating effect by ST, that is, rejecting the study hypothesis. CRs that differ according to the standardized path coefficients of the two groups are indicated in italics and underlined.

First, ST moderated the effect of PS (CR = 2.429) and IS (CR = 2.646) on RE. While these two paths were statistically significant in the high ST group, they were not significant in the low ST group, thereby supporting H6 and H7. Next, ST moderated the effect of RE on SA (CR = −2.773). In other words, the effect of RE on SA was greater in the low ST group compared to the high ST group, supporting H8. On the other hand, ST did not mediate the effect of RE on DA and PA, and thus, H9 and H10 were rejected.

## 5. Discussion

Our findings hold important implications for public health responses to viral risk. First, governments should aggressively provide products and information related to infection prevention and deliver messages to support and encourage public practice of infection-prevention activities. During times of viral risk, the government’s material, informational, and psychological support to the public have a direct impact on the public’s risk response efficacy and indirectly impacts their sanitary, distancing, and infection-prevention product purchasing activities. Further, support provided by the government is more effective than that provided by personal contacts such as families and friends. This is supported by Yan et al. [69], who showed that social support decreased depression and increased resilience in relation to the risk of virus infection. Furthermore, Labrague and De los Santos [70] mentioned that social support was associated with decreased levels of anxiety related to the COVID-19 pandemic and enhanced personal resilience.

Second, governments and all members of society must collaboratively respond to risks. The climate facilitated by such viral risk response translates to social trust, and high social trust improves the ability to respond to risk. Kuipers [71] and Roque et al. [72] reported that social trust in government institutions cannot prevent an individual from being affected by COVID-19, but it can help strengthen responses at all levels.

Third, risk communication that further emphasizes the importance of personal hygiene behaviors is needed. The impact of response efficacy on sanitation activities declined when social trust, the belief that all members of society will actively respond to the virus, was high compared to when it was low. This suggests that once a safe environment has been fostered through active responses against the virus by all members of society, individuals can become more complacent and lower their sanitation activities. According to the protection motivation theory [73], the public is motivated to engage in activities to protect themselves once they feel vulnerable to a risk. On the other hand, their motivation to engage in risk response activities is lost once they determine that the environment is safe. However, lowering the level of social trust because it diminishes sanitary activities may have an adverse impact on people’s ability to respond to a risk. Therefore, to promote sanitary activities in response to a viral risk, it is necessary to motivate the public to protect themselves, for which risk communication stressing the importance of personal hygiene behaviors during a viral risk is crucial. This is supported by our findings that social support provided by the government has both a direct and indirect impact on the public’s risk response efficacy and activities.

## 6. Limitations

This study has the following limitations. First, among many factors that impact the public’s risk response ability and activities, we only focused on the effects of social support and social trust. Subsequent studies should examine a more diverse set of variables. Second, the study population was confined to the Korean population. As a viral risk is a global health threat, it is important to conduct studies in different countries and cultures and comparatively analyze the results. Third, our use of an online survey generated sampling bias, as online surveys are only available for people who use the Internet.

## 7. Conclusions

This study identified the structure of social support, response efficacy, public activities, and social trust. In conclusion, governments must effectively and proactively provide material, informational, and psychological support, and social trust must be fostered through active response activities against the virus by all members of society in order to improve the public’s risk response ability and activities. This study shows that social support and social trust improve the public’s risk response ability and activities. Theoretically, the study findings will be conducive for understanding the public’s behavior in the face of viral risk. Furthermore, this study indicates that public support is more effective than individual support. Practically, the findings will serve as evidence for developing measures to overcome viral risk. These results might be used to organize the contents of virus infection-prevention campaigns for the public or to seek ways to actively prevent infection through material and psychological support.

## Figures and Tables

**Figure 1 ijerph-17-06589-f001:**
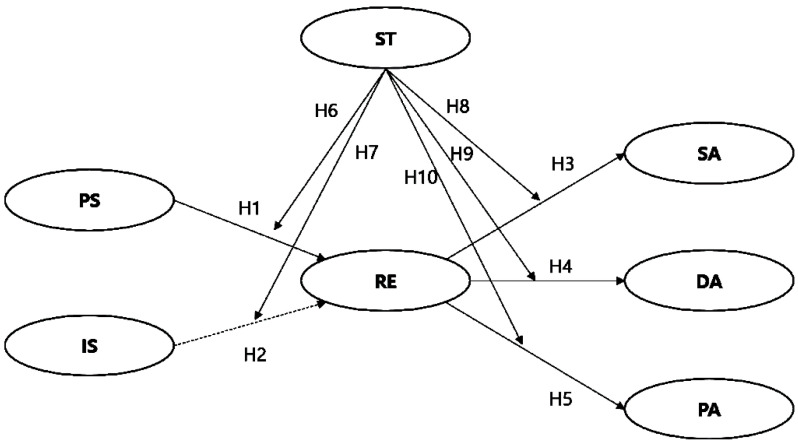
The hypothesized conceptual model. Note: DA, distancing activity; IS, individual support; PA, purchasing activity; PS, public support; RE, response efficacy; SA, sanitary activity; and ST, social trust.

**Figure 2 ijerph-17-06589-f002:**
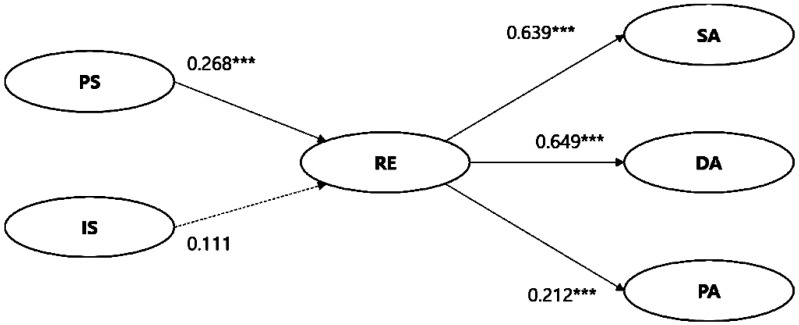
Structural model with standardized path coefficients. Note: *** *p* < 0.001. DA, distancing activity; IS, individual support; PA, purchasing activity; PS, public support; RE, response efficacy; SA, sanitary activity; and ST, social trust.

**Figure 3 ijerph-17-06589-f003:**
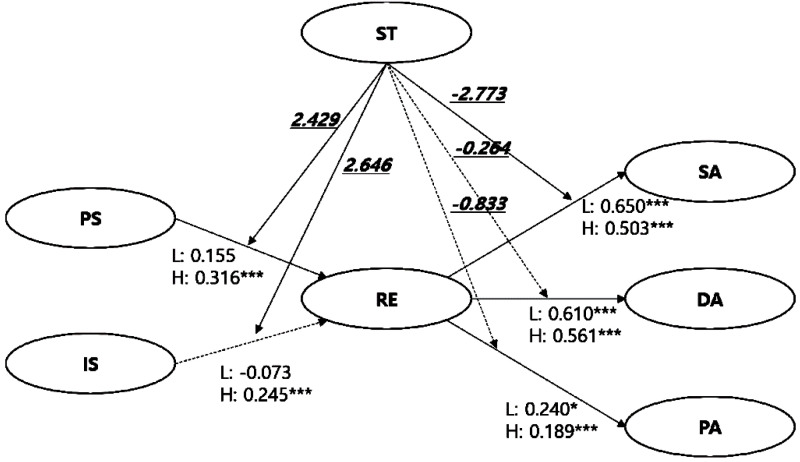
Structural model with standardized path coefficients and composite reliability between low and high groups on social trust. Note: * *p*<0.05, *** *p*<0.001. DA, distancing activity; IS, individual support; PA, purchasing activity; PS, public support; RE, response efficacy; SA, sanitary activity; and ST, social trust.

**Table 1 ijerph-17-06589-t001:** Descriptive statistics of survey participants (*n* = 620).

Characteristics	No	%
Gender		
Male	321	51.8
Female	299	48.2
Age group	M = 44.67	SD = 13.34
20–29	112	18.1
30–39	109	17.6
40–49	133	21.5
50–59	146	23.5
≥60	120	19.4
Education level		
High school or lower	124	20.0
Vocational school	100	16.1
Bachelor’s degree	343	55.3
Master’s degree and above	53	8.5
Monthly household income *	M = 477.96	SD = 317.36
Under 200	51	8.2
200–399	201	32.4
400–599	182	29.4
600–799	102	16.5
≥800	84	13.5
Area of residence		
Metropolitan area	297	47.9
Gyeongsang region	155	25.0
Jeolla-Jeju region	80	12.9
Gangwon-Chungchung region	88	14.2
Occupation		
Full-time employee	306	49.4
Part-time employee	34	5.5
Self-employed	72	11.6
Housewife	86	13.9
Student	44	7.1
Unemployed or other	78	12.6

Notes: * 10,000 South Korean won (USD1 = KRW 1185.00).

**Table 2 ijerph-17-06589-t002:** Constructs and Survey Questionnaire.

Construct 1: Public Support (PS)
PS1: My society provides me with infection-prevention products needed to overcome COVID-19.
PS2: My society provides me with information about infection prevention needed to overcome COVID-19.
PS3: My society provides me with emotional support (encouragement) to overcome COVID-19.
Construct 2: Individual support (IS)
IS1: People around me provide me with infection-prevention products needed to overcome COVID-19.
IS2: People around me provide me with information about infection prevention needed to overcome COVID-19.
IS3: People around me provide me with emotional support (encouragement) to overcome COVID-19.
Construct 3: Response efficacy (RE)
RE1: I know well how to protect my health from COVID-19.
RE2: I can control myself well to do things to protect my health from COVID-19.
RE3: I am willing to try to do things to protect myself from COVID-19.
Construct 4: Sanitation activity (SA)
SA1: To prevent COVID-19 infection, I always wear a mask when I leave my house to run errands.
SA2: To prevent COVID-19 infection, I wash my hands frequently.
SA3: To prevent COVID-19 infection, I cover my mouth and nose with my sleeve when I cough.
Construct 5: Distancing activity (DA)
DA1: To prevent COVID-19 infection, I refrain from outside activities and stay indoors.
DA2: To prevent COVID-19 infection, I practice physical distancing.
Construct 6: Infection-prevention product purchasing activity (PA)
BA1: To prevent COVID-19 infection, I bought facial masks.
BA2: To prevent COVID-19 infection, I bought antimicrobial and disinfecting products.
BA3: To prevent COVID-19 infection, I bought antimicrobial soap.
Construct 7: Social trust (ST)
ST1: I trust my government to strive to overcome COVID-19.
ST2: I trust our people to strive to overcome COVID-19.
ST3: I trust people around me to strive to overcome COVID-19.

**Table 3 ijerph-17-06589-t003:** Standardized estimate, reliability, and interconstruct correlations.

Construct	Measures	Standardized Estimate	Cronbach’s α	CR	AVE	Interconstruct Correlations							Mean (SD)
						PS	IS	RE	SA	DA	BA	ST	
PS	PS1	0.787	0.808	0.850	0.654	***0.809***							3.642 (0.790)
	PS2	0.791											
	PS3	0.864											
IS	IS1	0.761	0.877	0.889	0.728	0.693	***0.853***						3.528 (0.849)
	IS2	0.876											
	IS3	0.898											
RE	RE1	0.833	0.906	0.945	0.852	0.346	0.301	***0.923***					4.058 (0.685)
	RE2	0.887											
	RE3	0.898											
SA	SA1	0.855	0.833	0.930	0.827	0.279	0.169	0.639	***0.909***				4.215 (0.714)
	SA2	0.841											
	SA3	0.829											
DA	DA1	0.905	0.801	0.869	0.770	0.168	0.265	0.205	0.394	***0.877***			4.090 (0.715)
	DA2	0.740											
PA	BA1	0.733	0.802	0.793	0.562	0.213	0.187	0.649	0.726	0.339	***0.750***		3.776 (0.887)
	BA2	0.865											
	BA3	0.857											
ST	ST1	0.751	0.846	0.881	0.712	0.407	0.250	0.713	0.557	0.161	0.498	***0.844***	4.101 (0.761)
	ST2	0.885											
	ST3	0.786											

Notes: DA, distancing activity; IS, individual support; PA, purchasing activity; PS, public support; RE, response efficacy; SA, sanitary activity; and ST, social trust.

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
