# Peer review of "Impact of Social Support and Social Trust on Public Viral Risk Response: A COVID-19 Survey Study"

_ijerph, 2020, doi:10.3390/ijerph17186589_

Round 1

Reviewer 1 Report

The present study focuses on the impact of social support and social trust on the activities and efficacy of the public’s risk response in the case of COVID-19. It is an important point of view in times like the current pandemic, where social trust based on the information received is essential for society to take conscious and responsible measures. Despite that fact, the following elements should be reviewed:

  • Introduction: The introduction describes how the Covid occurred in South Korea, the measures, how the moments of risk affect people physically and mentally, but it is not justified what this study contributes with respect to others that have been carried out from this pandemic or from others.
  • Introduction: At the end of the introduction, the objectives of the study are described but it is also recommended to show the main conclusions of the study as well as the practical implications.
  • Theoretical Background and Hypothesis Development | 2.1 Social Support and Response Efficacy: Line 28 (Page 2)- the word disaster is repeated: "After a disaster, disaster victims mutually interact with various dimensions of the surrounding ..." perhaps it would be better to find a synonym or delete one of them, or structure the phrase in a different way.
  • Theoretical Background and Hypothesis Development | 2.3. Social Trust: Lines 28 and 29 (Page 4)- Hypotheses 6 and 7 are exactly the same:

            H6: Social trust will moderate the impact of public support on response efficacy.

H7: Social trust will moderate the impact of public support on response efficacy.

Could it be a transcription error? I would recommend a review of these hypotheses.

  • Methods | 3.1. Data Collection: Line 4 (Page 5)- The dates of the study surveys are: May 18-25, 2019, however, the Covid-19 pandemic began at the end of 2019 (December), and continues in 2020 (since January), so I do not understand that period of analysis since the existence of COVID-19 was unknown.
  • Methods | 3.2. Measures | 3.2.1 Social support: Page 6, Table 2: This phrase appears in bold in the table and the rest do not, perhaps criteria should be unified: “Construct1: Public support (PS)”
  • Conclusion: In the conclusion, I would recommend further detailing the practical implications of this study as well as the scientific contributions with respect to others that have been carried out.

Author Response

Response to Comments from Reviewer 1 and Revisions

â–ˇ Comment 1

Reviewer’s comment:

Introduction:

The introduction describes how the Covid occurred in South Korea, the measures, how the moments of risk affect people physically and mentally, but it is not justified what this study contributes with respect to others that have been carried out from this pandemic or from others.

At the end of the introduction, the objectives of the study are described but it is also recommended to show the main conclusions of the study as well as the practical implications.

Answer: We (all of the authors) appreciate your constructive feedback. We have inserted the contributions of this study as follows (please see page 2, lines 32-37):

“This study aimed to show how governments must effectively and proactively provide material, informational, and psychological support and that social trust must be fostered through active response activities against the virus by all members of society in order to improve the public’s risk response ability and activities. Theoretically, the study findings will be conducive to understanding the public’s behavior in the face of viral risk, and practically, the findings will serve as evidence for developing measures to overcome such risk.”

â–ˇ Comment 2

·         Reviewer’s comment:

·         Theoretical Background and Hypothesis Development | 2.1 Social Support and Response Efficacy:

·         Line 28 (Page 2)- the word disaster is repeated:

"After a disaster, disaster victims mutually interact with various dimensions of the surrounding ..." perhaps it would be better to find a synonym or delete one of them, or structure the phrase in a different way.

Answer: Thank you for your comment. We have revised the sentence as follows (please see page 2, line 40):

“After a disaster, victims mutually interact with various dimensions..."

â–ˇ Comment 3

Reviewer’s comment:

·         Theoretical Background and Hypothesis Development | 2.3. Social Trust: Lines 28 and 29 (Page 4)- Hypotheses 6 and 7 are exactly the same

            H6: Social trust will moderate the impact of public support on response efficacy.

H7: Social trust will moderate the impact of public support on response efficacy.

Could it be a transcription error? I would recommend a review of these hypotheses.

Answer: Yes, you are right. This was our mistake. We appreciate you giving us an opportunity to revise the text. Hypotheses 7 was revised from public support to individual support (please see page 4, line 40):

“H7: Social trust will moderate the impact of individual support on response efficacy.”

â–ˇ Comment 4

Reviewer’s comment:

Methods | 3.1. Data Collection: Line 4 (Page 5)- The dates of the study surveys are: May 18-25, 2019, however, the Covid-19 pandemic began at the end of 2019 (December), and continues in 2020 (since January), so I do not understand that period of analysis since the existence of COVID-19 was unknown.

Answer: Yes, you are right. The year 2019 was a mistake. We apologize for the confusion; we have revised the year, from 2019 to 2020 (please see page 5, line 12).

â–ˇ Comment 5

Reviewer’s comment:

Methods | 3.2. Measures | 3.2.1 Social support: Page 6, Table 2: This phrase appears in bold in the table and the rest do not, perhaps criteria should be unified: “Construct1: Public support (PS)”

Answer: Thank you for your highly detailed review. We unified the phrase by modifying the bold font. Thank you for your comment. Please see Table 2.

â–ˇ Comment 6

Reviewer’s comment:

Conclusion: In the conclusion, I would recommend further detailing the practical implications of this study as well as the scientific contributions with respect to others that have been carried out.

Answer: Based on your opinion, we have revised most of the conclusion (please see page 11, lines 32-37).

“This study shows that social support and social trust improve the public’s risk response ability and activities. Theoretically, the study findings will be conducive for understanding the public’s behavior in the face of viral risk. Furthermore, this study indicates that public support is more effective than individual support. Practically, the findings will serve as evidence for developing measures to overcome viral risk. These results might be used to organize contents of virus infection prevention campaigns for the public or to seek ways to actively prevent infection through material and psychological support.”

++We appreciate the reviewer’s excellent comments, which helped us improve the quality of our manuscript. Thank you! ++

Reviewer 2 Report

This study examined the role of social support and social trust on preventive behaviors and response efficacy in the context of COVID-19. This study contributes novel findings to the control of the current COVID-19 pandemic. However, I find this manuscript is a bit disorganized and lack of clarity. Here are my suggestions:

Introduction

Page 1, line 38-42, please provide the confirmed case number so the readers have a better idea about what you mean by "skyrocket after the diagnosis of the 31st patient" and "additional diagnoses had dropped significantly from April 14, 2020", and "a second round of CVOID-19 infections in Seoul". 

Page 2, line 6, please clarify the term "public masks". 

Page 2, line 16, "Another important factor is trust". You mean social trust correct? Please provide a definition of social trust. 

Page 2, line 39-40, "In addition, Babcicky and Seebauer also distinguished between sources of social support." Did they classified social support into individual and public support? 

Page 3, line 15, "In summary, social support can be classified as public support and individual support". I did not see a clear rationale for such classification in the previous paragraphs.

Methods

Data Collection

How did the professional online survey company recruit potential participants? Did the company have a pool of registered users? How many are they? How many received study invitation and how many complete the survey (response rate)? Did the participants receive any incentives after they complete the online survey? 

Measures

The measures under "3.2.1 Social support" were go beyond social support. You included sanitation activity, distancing activity, infection prevention product purchasing activity...under that subtitle. Please create additional subtitles to describe these measures.

Analysis

Page 7, line 17, "The reliability of the scale was test with confirmatory factor analysis (CFA)". CFA examines validity instead of reliability. 

Page 7, line 21, what does "dl" mean?

Page 7, line 18, "Model fit computed from CFA was interpreted per the criteria proposed by...". Please list the criteria. What are the parameters cut of points for models considered as good fit, acceptable fit, and poor fit? 

Similarly, page 7, line 22, "the criteria suggested by Fornell and Larchker...", please list the criteria. 

Page 7, line 24, what is the rationale to use Woo's method when testing moderation effects? How does it better than the traditional moderation analysis? 

Results

Please spell out the construct names in the Figures or add a figure note as what you did for Table 3. 

Discussion

The discussion section is insufficient. Please link your findings with previous studies and future interventions. 

Limitations

There are other limitations for this study. For example, the cross-sectional design of the study cannot identify causal inferences. And the online survey design generates sampling bias. 

Author Response

Response to Comments from Reviewer 2 and Revisions

â–ˇ Comment 1

Reviewer’s comment:

Introduction: Page 1, line 38-42, please provide the confirmed case number so the readers have a better idea about what you mean by "skyrocket after the diagnosis of the 31st patient" and "additional diagnoses had dropped significantly from April 14, 2020", and "a second round of CVOID-19 infections in Seoul". 

Answer: We (all of the authors) appreciate your feedback. We revised this part as follows (please see page 1–2, lines 37–2):

“However, the number of confirmed cases began to skyrocket after the diagnosis of the 31st patient on February 17, 2020. According to the South Korean central government and the Korea Centers for Disease Control and Prevention (KCDC) report [2], an average of 207.4 additional confirmed cases per day were reported from February 17, 2020, to April 13, 2020. On February 29, 2020, an additional 1,128 cases were confirmed to be infected with COVID-19. Due to aggressive management and prevention measures implemented by the KCDC, the rate of additional diagnoses dropped significantly after April 14, 2020, to an average of less than 30 cases per day. However, in June 2020, South Korean authorities announced the potential need for stricter physical measures should the increasing infection rate not subside [3,4].”

â–ˇ Comment 2

Reviewer’s comment:

Introduction :Page 2, line 6, please clarify the term "public masks". 

Answer: We revised this part as follows (please see page 2, lines 7–10):

“These measures include the provision of necessary equipment, such as the provision of “public masks,” masks provided by designated pharmacies for a low price in response to the soaring prices of facial masks, and the order of a supply of hand sanitizers by major hospitality facilities”

â–ˇ Comment 3

Reviewer’s comment:

Introduction: Page 2, line 16, "Another important factor is trust". You mean social trust correct? Please provide a definition of social trust. 

Answer: We revised this part as follows (please see page 2, lines 18–24):

“Another important factor is social trust. Social trust is based on cultural values and refers to decisions about whether to trust or distrust anonymous others (people and institutions) [10]. It provides psychological stability when individuals interact with others and is therefore of great import in increasingly atomistic modern societies [11]. Social trust impacts a person’s overall perception of risk and their psychological response and behavior related to the risk [12,13], as well as effectively promoting risk recovery [14,15].”

+++References+++

[10] Earle, T.C.; Cvetkovich, G. Social trust and culture in risk management. In Social Trust and the Management of Risk; Cvetkovich, G., Lofstedt, R., Eds.; Routledge: London, UK, 1999; pp. 9–21.

[11] Sønderskov, K.M.; Dinesen, P.T. Trusting the state, trusting each other? The effect of institutional trust on social trust. Polit Behav 2016, 38, 179–202.

â–ˇ Comment 4

Reviewer’s comment:

Theoretical Background and Hypothesis Development | 2.1. Social Support and Response Efficacy:

Page 2, line 39-40, "In addition, Babcicky and Seebauer also distinguished between sources of social support." Did they classified social support into individual and public support? 

Answer: Yes, you are right. To clarify the meaning of this part, the sentence was deleted, and a citation was added to the previous sentence as follows (please see pages 2–3, lines 49–2):

“and classification by provider of support, namely individual support provided by personal contacts, including family and friends, and public support provided by government and local communities [18,21, 22]. In this study, we classified social support by provider type (individual and public support) and examined its relationship with response efficacy.”

â–ˇ Comment 5

Reviewer’s comment:

Theoretical Background and Hypothesis Development | 2.1. Social Support and Response Efficacy:

Page 3, line 15, “In summary, social support can be classified as public support and individual support”. I did not see a clear rationale for such classification in the previous paragraphs.

Answer: Please see page 2, lines 47–48 and page 3, lines 8–16. We mentioned that social support can be classified as public or individual support. Furthermore, to clarify the relationship between public/individual support and response efficacy, we checked related citations and revised some of the text as follows (please see page 3, lines 12–20):

“Duncan and McAuley [25] and Resnick et al. [26] reported that social support, including public and individual support, indirectly boosts exercise behaviors through efficacy in older adults. Berkman [27] found that peer and colleague support increases individuals’ willingness to exercise, thereby demonstrating that individual support has a positive effect on efficacy. Wind et al. [28] found that high levels of cognitive social support lead to better mental health outcomes following a disaster. Further, Bashirian et al. [29] reported that social support affects breast self-examination behavior, implying that social support can boost self-efficacy. Social support also has an important impact on risk. Yoo and Joo [30] and Kim et al. [31] revealed that public information support regarding a risk impacts response efficacy:”

â–ˇ Comment 6

Reviewer’s comment:

Methods | Data Collection

How did the professional online survey company recruit potential participants? Did the company have a pool of registered users? How many are they? How many received study invitation and how many complete the survey (response rate)? Did the participants receive any incentives after they complete the online survey? 

Answer: Thank you for your comment. We have revised the Data collection section with reference to your comments as follows (please see page 5, lines 11–19):

“Study data were collected using a questionnaire administered by a professional online survey company, Macomill Embrain, for eight days (May 18–25, 2020). The company has a panel of 1,249,392 adults aged 20 or older, which is equivalent to 3.5% of the South Korean population. The questionnaire was converted to an online-based survey to make online administration easier and was distributed to a total of 7,320 panelists via email. One panelist received no more than two response invitations. Prior to beginning the survey, participants were asked whether they voluntarily provided their consent to participate, and only those who indicated their voluntary consent were allowed to participate in the survey. The panelists that completed the online survey received incentives equivalent to about $3. The final response rate was 8.5%.”

â–ˇ Comment 7

Reviewer’s comment:

Methods | Measures

The measures under "3.2.1 Social support" were go beyond social support. You included sanitation activity, distancing activity, infection prevention product purchasing activity...under that subtitle. Please create additional subtitles to describe these measures.

Answer: Yes, we fully agree with you. We have briefly mentioned the six items at the beginning of 3.2 Measures as follows (please see page 6, lines 2–5):

“3.2 Measures

In this study, the following six items were developed: public support (PS), individual support (IS), response efficacy (RE), sanitation activity (SA), distancing activity (DA), infection-prevention product purchasing activity (PA), and social trust (ST) (Table 2).”

â–ˇ Comment 8

Reviewer’s comment:

Methods | Analysis

Page 7, line 17, "The reliability of the scale was test with confirmatory factor analysis (CFA)". CFA examines validity instead of reliability. 

Answer: Yes, you are right. The sentence was revised as follows (please see page 7, lines 24-25):

“The validity of the scale was tested with confirmatory factor analysis (CFA) and reliability analysis.”

â–ˇ Comment 9

Reviewer’s comment:

Methods | Analysis

Page 7, line 21, what does "dl" mean?

Answer: We apologize; “dl” was mistakenly inserted. We appreciate you giving us an opportunity to revise this. The sentence was revised as follows (please see page 7, lines 30–32):

“Further, the reliability of the scale was tested by comparing the standardized and variance estimates computed from CFA, composite reliability (CR), average variance extracted (AVE), square roots of AVEs, and Cronbach’s α. ”

â–ˇ Comment 10

Reviewer’s comment:

Methods | Analysis

Page 7, line 18, "Model fit computed from CFA was interpreted per the criteria proposed by...". Please list the criteria. What are the parameters cut of points for models considered as good fit, acceptable fit, and poor fit? 

Similarly, page 7, line 22, "the criteria suggested by Fornell and Larchker...", please list the criteria. 

Answer: We have revised this part as follows (please see page 7, lines 24–34):

“The validity of the scale was tested with confirmatory factor analysis (CFA) and reliability analysis. This study determines the fitness of the model through the following criteria: the Chi-square minimum/degree of freedom (CMIN/df), the root mean-squared residual (RMR), the root-mean-squared error of approximation (RMSEA), the goodness-of-fit index (GFI), the adjusted-goodness-of-fit index (AGFI), and the comparative-fit index (CFI). For a model with acceptable quality, the following threshold values are recommended: CMIN/df ≤ 5, RMR ≤ 0.05, RMSEA ≤ 0.08, GFI ≤ 0.9, AGFI ≥ 0.9, CFI ≤ 0.9 [59–62]. Further, the reliability of the scale was tested by comparing the standardized and variance estimates computed from CFA, composite reliability (CR), average variance extracted (AVE), square roots of AVEs, and Cronbach’s α. For a model with acceptable quality, the following threshold values are recommended: CR ≥ 0.7, AVE ≥ 0.5, Cronbach’s α ≥ 0.7, square roots of AVEs ≥ of the interconstruct correlation coefficient [63–64]. ”

+++References+++

[62] Balong, A. Acceptance of e-learning systems: A serial multiple meditation analysis. Stud Inform Control 2015, 24, 101–110.

.

â–ˇ Comment 11

Reviewer’s comment:

Methods | Analysis

Page 7, line 24, what is the rationale to use Woo's method when testing moderation effects? How does it better than the traditional moderation analysis? 

Answer: The test method we used is one of several testing moderation methods that is used in the literature and is not only Woo’s. To clarify this, we have added two citations that use the same method (please see page 7, lines 36–37):

+++References+++

[66] Harish Sujan, B.A.; Weitz, N.K. Learning orientation, working smart, and effective selling. J Mark 1994, 58(3), 39–52.

[68] Lee, H.S.; Lim, J.H. Structural Equation Modeling with AMOS 6.0. Bobmunsa Publishing Company: Paju, Korea, 2007.

â–ˇ Comment 12

Reviewer’s comment:

Results: Please spell out the construct names in the Figures or add a figure note as what you did for Table 3. 

Answer: I added all construct names to Figures 1, 2, and 3. Please see the note in all of the figures.

Figure 1: Note: DA, distancing activity; IS, individual support; PA, purchasing activity; PS, public support; RE, response efficacy; SA, sanitary activity; and ST, social trust.

Figure 2 and 3: Note: *p<0.05, **p<0.01, ***p<0.001. DA, distancing activity; IS, individual support; PA, purchasing activity; PS, public support; RE, response efficacy; SA, sanitary activity; and ST, social trust.

â–ˇ Comment 13

Reviewer’s comment:

Discussion:

The discussion section is insufficient. Please link your findings with previous studies and future interventions. 

Answer: I have added previous studies as follows (please see page 10, lines 27-30 and lines 33-35):

“This is supported by Yan et al. [69], who showed that social support decreased depression and increased resilience in relation to the risk of virus infection. Furthermore, Labrague and De los Santos [70] mentioned that social support was associated with decreased levels of anxiety related to the COVID-19 pandemic and enhanced personal resilience.”

“Kuipers [71] and Roque et al. [72] reported that social trust in government institutions cannot prevent an individual from being affected by COVID-19, but it can help strengthen responses at all levels.”

+++References+++

[69] Yan, H.; Li, X.;, Li, J.; Wang, W.; Yang, Y.; Yao, X.; Yang, N.; Li, S. Association between perceived HIV stigma, social support, resilience, self-esteem, and depressive symptoms among HIV-positive men who have sex with men (MSM) in Nanjing, China. Psychol Sociomed Aspects AIDS/HIV 2019, 31(9), 1069–1976.

[70] Labrague, L.J.; De los Santos, J.A.A. COVID-19 anxiety among front-line nurses: predictive role of organizational support, personal resilience and social support. J Nurs Manag 2020, 00, 1–9. https://doi,org/10.1111/jonm.13121

[71] Kuipers, S. Editorial: Sanity and resilience in times of Corona. Risk Hazards Crisis Public Policy 2020, 11(2), 110–115.

[72] Roque, A.D.; Pijawka, D.; Wutich, A. The role of social capital in resiliency: disaster recovery in Puerto Rico. Risk Hazards Crisis Public Policy 2020, 11(2), 204–235.

â–ˇ Comment 14

Reviewer’s comment:

Limitations

There are other limitations for this study. For example, the cross-sectional design of the study cannot identify causal inferences. And the online survey design generates sampling bias.

Answer: Yes, you are right. Based on your opinion, I have inserted the following sentence (please see page 11, lines 21-23):

“Third, our use of an online survey generated sampling bias, as online surveys are only available for people who use the Internet.”

â–ˇ Comment 15

Reviewer’s comment:

English language and style: Moderate English changes required

Answer: We have checked the manuscript thoroughly and it has been further reviewed by Editage, which is a professional English language editing company. All of the revisions are highlighted in yellow throughout the manuscript.

++We appreciate the reviewer’s excellent comments, which helped us improve the quality of our manuscript. Thank you! ++

Round 2

Reviewer 2 Report

Thanks for your revision. All my comments have been addressed.